# A Thermomechanical Coupling Constitutive Model of Concrete Including Elastoplastic Damage

**Liang Li** [1,*], **Hongwei Wang** [1], **Jun Wu** [2,*] and **Wenhua Jiang** [1]

1   Key Laboratory of Urban Security and Disaster Engineering, Beijing University of Technology, Ministry of Education, Beijing 100124, China; 18811169721@163.com (H.W.); jiangwenhua@vtjz.com (W.J.)
2   School of Urban Railway Transportation, Shanghai University of Engineering Science, Shanghai 201620, China
*   Correspondence: liliang@bjut.edu.cn (L.L.); cvewujun@sues.edu.cn (J.W.); Tel.: +86-10-6739-2430 (L.L.)

**Abstract:** The thermomechanical coupling constitutive model of concrete is a critical subject for the theoretical investigation and numerical simulation of the mechanical behaviors of concrete members and structures at high temperature. This paper presents a thermomechanical coupling constitutive model for the description of the mechanical behaviors of concrete at different temperatures. The expression of the elastic strain increment is derived with the free energy function including the temperature variable. The expression of the plastic strain increment is derived from the yield function based on the Drucker–Prager strength criterion. The elastoplastic damage effect is included in this constitutive model. The damage variable is included in the yield function to consider the effect of the damage on the elastoplastic mechanical behaviors of concrete. The proposed constitutive model is validated by the comparison of the simulation results of the uniaxial compression tests of concrete at different temperatures with the corresponding test results. The simulation results accord well with the test results at different temperatures. This indicates that the proposed constitutive model can characterize the mechanical behaviors of concrete at different temperatures with considerable accuracy. The proposed constitutive model was applied to simulate an axially compressive concrete column. The simulation results are consistent with the essential mechanical response behaviors of concrete members at different temperatures.

**Keywords:** concrete; thermomechanical coupling; constitutive model; elastoplastic damage; thermodynamics

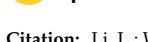



## 1. Introduction

Concrete is commonly used in some engineering structures that are exposed to high temperature, e.g., industry boilers, chimneys and nuclear reactor shields. Fires or blasts can also cause high temperature and subsequently serious damage to the structures [1–5], potentially resulting in a large number of casualties and significant economic losses. For example, on 9 February 2009, a serious fire occurred in the new building of the Television Culture Center of China Central Television (CCTV). It caused a direct economic loss of 160 million yuan. A fire at a 28-story apartment on Jiaozhou Road of Jing'an District in Shanghai on 15 November 2011 caused 58 deaths and 71 injuries, with a direct economic loss of 158 million yuan. The mechanical properties of concrete at elevated temperature need to be studied thoroughly for fire-resistant design and assessment of concrete structures and members.

The behaviors of concrete at elevated temperature are generally investigated by experiments. Guo and Li [6] conducted concrete tests of temperature loading and stress loading and proposed a thermomechanical coupling constitutive model of concrete. Qin et al. [7] conducted biaxial tension–compression tests of concrete under four stress ratios after high temperature exposure using a large-scale static–dynamic triaxial test system. A failure criterion was established in the principal stress space and octahedral stress space.

Zhang et al. [8] conducted triaxial equal proportion compressive tests to analyze the influence of stress ratio and temperature on the triaxial compressive strength of concrete. Test results show that triaxial compressive strength is considerably larger than uniaxial strength for the same designated temperature, and middle principal stress has significant influence on both triaxial compressive strength and strain at peak stress. Liang et al. [9] carried out a compressive strength test on concrete cube specimens after high temperature exposure, considering the effects of temperatures, cooling methods, aggregate types, standing time and strength grades. Jia et al. [10] carried out an experiment to present the influences of various cooling methods and standing times on the residential strength of concrete after elevated temperature exposure. Xing et al. [11] proposed a calculation model for the short-term thermal creep of concrete based on the experimental results. The model can be used to analyze the mechanical performance of concrete structures subjected to high temperature. Yu et al. [12] conducted a series of experiments on cube specimens to study the relationship between concrete compressive strength and some influence factors at different temperatures and fire durations. Shao et al. [13] studied the influence of different temperatures and heating times on the compressive strength, elastic modulus and stress–strain relationship of concrete with a hydraulic servo testing system. Kizilkanat et al. [14] investigated the effect of high temperature on the thermal conductivity and moisture resistance factor of concrete with different types of pozzolans and aggregates. Ergün et al. [15] studied the effects of elevated temperatures and cement dosages on the mechanical properties of concrete. Zhai et al. [16] carried out uniaxial compression tests on C35 concrete under normal and high temperatures to investigate the deterioration of the mechanical properties of concrete caused by the high temperatures. Tang [17] examined the effects of individual and mixed fiber on the mechanical properties of lightweight aggregate concrete (LWC) after exposure to elevated temperatures. Głowacki and Kowalski [18] presented the results of tests for possible stiffness changes in bent reinforced concrete elements exposed to simultaneous action of load and high temperature. Ma et al. [19] and Zhang et al. [20] presented reviews of the mechanical properties of normal concrete and steel-fiber-reinforced concrete at high temperature, respectively. The compressive behaviors of concrete are heavily conditioned by passive confinement offered by transverse reinforcements, consisting of internal hoops and additional external strengthening, such as fiber reinforced polymer (FRP) wraps and steel jackets [21,22].

When the mechanical properties of concrete under high temperature conditions are studied by theorical analysis, a thermomechanical coupling constitutive model should be used. The constitutive models developed based on the classical plasticity theory can usually satisfy the stability postulate (e.g., Drucker's stability postulate and Iliushin's postulate). Gernay et al. [23] developed a multiaxial constitutive model based on the combination of elastoplasticity and damage theories. The isotropic damage was assumed. This model was applied to the numerical simulation of concrete structural members of a fire situation [24]. Yu et al. [25] proposed a concrete constitutive model considering coupled effects of high temperature and high strain rate by modifying the Drucker–Prager constitutive model. Bouras et al. [26] presented a non-linear thermo-viscoelastic rheological model based on fractional derivatives for high temperature creep in concrete. Liang et al. [27] established a dynamic damage constitutive model combining statistical damage and a viscoelastic model for the description of the dynamic mechanical properties of basalt-fiber-reinforced concrete exposed to different temperatures.

The thermomechanical coupling constitutive models based on the classical plasticity theory may not strictly satisfy the laws of thermodynamics. The development of plastic or irrecoverable strains of concrete exhibits features that are foreign to the classical theory of thermodynamics. A thermomechanical coupling constitutive model should be developed based on thermodynamics and consider the strictness of the theory foundation. Heinfling et al. [28] proposed a constitutive model concerning concrete cracks based on the theory of thermodynamics. Ju and Zhang [29] presented energy-based coupled elastoplastic damage theories. The proposed formulation employs irreversible thermodynamics and

internal state variable theory for ductile and brittle materials. Gawin et al. [30] modeled a computational analysis of hygro-thermal and mechanical behavior of concrete structures at high temperature including the heat and mass transfer processes. Nechnech et al. [31] built a computational model for the thermomechanical analysis of concrete structures at high temperature using the finite element method. Wu and Li [32] defined the total Helmhotlz free energy and the elastoplastic damage energy release rate based on the damage mechanism of concrete and developed a series of elastoplastic damage constitutive models that satisfy the principles of thermodynamics. Pont et al. [33] proposed a multiphase thermo-hydro-mechanical model for concrete at high temperature. Krairi and Doghri [34] proposed a thermodynamically based constitutive model for isotropic homogeneous thermoplastic polymers under arbitrary multiaxial and non-monotonic loadings. Torelli et al. [35] developed a confinement-dependent load-induced thermal strain constitutive model for concrete subjected to temperatures up to 500 °C. Chen et al. [36] proposed a stress–strain model to describe the residual tensile stress–strain relationship of concrete considering the effect of cyclic damage and high temperature. Pečenko et al. [37] presented a two-phase computational model for determining the response of prestressed hollow-core concrete slab exposed to natural fire. Yao et al. [38] developed an elastoplastic damage constitutive model of concrete considering the effects of dehydration and pore pressure at high temperatures. Torelli et al. [39] presented a moisture-dependent thermomechanical constitutive model for concrete able to capture the effect of the moisture content of the material on its mechanical behavior under compressive loads and high transient temperatures.

Under the combined action of loads and high temperature, the mechanical properties of concrete deteriorate due to the growth and accumulation of inner cracks. This can be described by the damage evolution. A thermomechanical coupling constitutive model should characterize the damage evolution of concrete under the combined action of high temperature and loads. On the other hand, the present constitutive models can describe physical and mechanical behaviors of concrete such as thermal creep and crack, but they have complex mathematical expressions and include numerous parameters. It is not convenient to use these models for practical problems.

In this study, a thermomechanical coupling constitutive model including elastoplastic damage was developed to characterize the mechanical behaviors of concrete under high temperature conditions. The effect of temperature on the material performance is described in a relatively simple way. The temperature variable is included directly in the expressions of the yield function and hardening parameter to characterize the effect of temperature on the elastoplastic mechanical behaviors of concrete. The constitutive model proposed in the current study has relatively simple expressions. Fewer parameters are included in this model, and they all have explicit physical meanings. The developed constitutive model is validated by the comparison of numerical simulation results of the model with the corresponding experimental results. The model was also applied to the numerical simulation of a concrete member under high temperature conditions.

The structure of the paper is as follows: After the introduction, Section 2 presents the derivation process of the proposed constitutive model in detail. The expressions of the model are presented in this section. Section 3 outlines the validation of the proposed constitutive model. The model is validated by the comparison of the numerical simulation results of the model with the test results. Section 4 presents the application of the proposed model. It was applied to the numerical simulation of a concrete member. Lastly, in Section 5, the conclusions are presented.

## 2. Model Development

In the constitutive model developed in the current study, the expression of the elastic strain increment is derived based on the free energy function including the temperature variable. The expression of the plastic strain increment is derived based on the classical plastic theory. The yield function including damage variable is derived from the Drucker–Prager strength criterion. The temperature variable is included in the expression of the

hardening parameter to consider the effect of temperature on the material hardening behavior. The evolution equation of the damage variable was developed based on the Najar damage theory. The expressions of the constitutive model can be obtained by the sum of elastic strain increment and plastic strain increment.

### 2.1. Elastic Stress–Strain Model

The uncoupling assumption of elasticity and plasticity for concrete was adopted for the development of the elastic stress–strain model. The elasticity and plasticity are unattached and do not impact each other. The free energy function consists of thermoelastic free energy and thermoplastic free energy. Based on the above assumption, the free energy function can be written as follows

$$\psi\left(\varepsilon_{ij}^e, \varepsilon_d^p, T, D\right) = \psi^e\left(\varepsilon_{ij}^e, T\right) + \psi^p\left(\varepsilon_d^p, T, D\right) \tag{1}$$

where $\psi$ is the Helmholtz free energy function, $\psi^e$ is the elastic Helmholtz free energy function and $\psi^p$ is the plastic Helmholtz free energy function. $\varepsilon_{ij}^e$ is the elastic strain tensor, $\varepsilon_d^p$ is the plastic shear strain, $T$ is the temperature and $D$ is the damage variable.

The thermodynamic law of the isothermal pure mechanical process can be expressed as follows

$$\dot{\eta} = \sigma : \dot{\varepsilon} - \dot{\psi} \geq 0 \tag{2}$$

The derivative of the free energy function is

$$\dot{\psi} = \frac{\partial \psi^e}{\partial \varepsilon_{ij}^e}\dot{\varepsilon}_{ij}^e + \frac{\partial \psi}{\partial T}\dot{T} + \frac{\partial \psi}{\partial D}\dot{D} + \frac{\partial \psi^P}{\partial \varepsilon_d^P}\dot{\varepsilon}_d^p \tag{3}$$

Substituting Equation (3) into Equation (2), we can get

$$\left(\sigma_{ij} - \frac{\partial \psi^e}{\partial \varepsilon_{ij}^e}\right)\dot{\varepsilon}_{ij}^e - \frac{\partial \psi}{\partial T}\dot{T} - \frac{\partial \psi}{\partial D}\dot{D} + \left(\sigma_{ij} : \dot{\varepsilon}_{ij}^p - \frac{\partial \psi^p}{\partial \varepsilon_d^p}\dot{\varepsilon}_d^p\right) \geq 0 \tag{4}$$

In order to let Equation (4) be satisfied for arbitrary $\varepsilon_{ij}^e$, we use

$$\sigma_{ij} = \frac{\partial \psi^e}{\partial \varepsilon_{ij}^e} = \frac{\partial \psi^e\left(\varepsilon_{ij}^e, T\right)}{\partial \varepsilon_{ij}^e} \tag{5}$$

An elastic free energy function proposed by Stabler and Baker [40] was adopted. It includes both the mechanical damage variable and temperature variable. It can be written as follows

$$\psi^e = \frac{1}{2}\varepsilon_{ij}^e C_{ijkl}^e \varepsilon_{kl}^e - \vartheta(T - T_0)\varepsilon_{ij}^e + c_k\left[T - T_0 - T\ln\frac{T}{T_0}\right] \tag{6}$$

where $\vartheta$ is the thermomechanical coupling tensor, $\vartheta = C_{ijkl}^e \beta_{kl}$; $\beta_{kl}$ is the thermal expansion coefficient tensor, $\beta_{kl} = \beta\delta_{kl}$ for the isotropic case and $\delta_{kl}$ is the Kronecker $\delta$. $C_{ijkl}^e$ is the elastic modulus tensor in the damaged state, $C_{ijkl}^e = (1 - D)C_{ijkl}^0$; $C_{ijkl}^0$ is the initial elastic modulus tensor in the undamaged state. $T_0$ is the initial temperature, $T_0 = 20\,°C$. $c_k$ is the specific heat capacity, and it is assumed to be constant and independent of strain (or stress) and temperature.

Substituting Equation (6) into Equation (5), we can get

$$\sigma_{ij} = C_{ijkl}^e \varepsilon_{kl}^e - (T - T_0)C_{ijkl}^e \beta\delta_{kl} \tag{7}$$

The incremental elastic stress–strain model can be expressed as

$$d\sigma_{ij} = C_{ijkl}^e d\varepsilon_{kl}^e \tag{8}$$

### 2.2. Plastic Stress–Strain Model

The yield function is derived based on the Drucker–Prager strength criterion [41], and it can be written as

$$F = \alpha I_1 + \sqrt{J_2} - (1 - D)K \tag{9}$$

where $I_1$ is the first stress tensor invariant, $I_1 = \sigma_{kk}$. $J_2$ is the second deviatoric stress tensor invariant, $J_2 = 1/2 s_{ij}s_{ij}$. $\alpha$ is the parameter of the Drucker–Prager strength criterion and can take a value of 0.01 [41]. $K$ is the hardening parameter. The damage variable $D$ is included in the yield function to consider the effect of damage on the elastoplastic mechanical behaviors of concrete.

When the effect of temperature on the plastic mechanical behavior of concrete is considered and the plastic shear strain is taken as the inner variable, the hardening parameter can be expressed as

$$K = K(\varepsilon_d^p, T) = \frac{\varepsilon_d^p}{A + B\varepsilon_d^p} \tag{10}$$

where $\varepsilon_d^p$ is the plastic shear strain and $A$ and $B$ are model parameters, which can be expressed as

$$A = \frac{1}{G^T}, \ \frac{1}{B} = \alpha f_c^T + f_c^T \tag{11}$$

where $G^T$ and $f_c^T$ are the shear modulus and compressive strength of concrete at the temperature $T$, respectively. They can be written as

$$G^T = \frac{E_{(T)}}{2[1 + \nu_{(T)}]}, \ f_c^T = \frac{f_c}{1 + 2.4(T - 20)^6 \times 10^{-17}} \tag{12}$$

where $E_{(T)}$ and $\nu_{(T)}$ are the elastic modulus and Poisson's ratio of concrete at the temperature $T$ and $f_c$ is the compressive strength of concrete under normal temperature conditions.

According to the consistency condition, we obtain

$$dF = \frac{\partial F}{\partial \sigma_{ij}}d\sigma_{ij} + \frac{\partial F}{\partial K}\frac{\partial K}{\partial \varepsilon_{ij}^p}d\varepsilon_{ij}^p = 0 \tag{13}$$

From the orthogonal flow rule, we can get the expression of the plastic strain increment as follows

$$d\varepsilon_{ij}^p = d\lambda \frac{\partial F}{\partial \sigma_{ij}} \tag{14}$$

where $d\lambda$ is a scalar multiplier. The plastic shear strain increment can be expressed as follows

$$d\varepsilon_d^p = \left(\frac{2}{3}d\varepsilon_{mn}^p : d\varepsilon_{mn}^p\right)^{\frac{1}{2}} \tag{15}$$

From Equations (14) and (15), we can get

$$d\varepsilon_d^p = d\lambda \left(\frac{2}{3}\frac{\partial F}{\partial \sigma_{mn}} : \frac{\partial F}{\partial \sigma_{mn}}\right)^{\frac{1}{2}} \tag{16}$$

From Equations (13) and (14), we can get

$$d\sigma_{ij} = \frac{-\frac{\partial F}{\partial K}\frac{\partial K}{\partial \varepsilon_{ij}^p}d\lambda \frac{\partial F}{\partial \sigma_{ij}}}{\frac{\partial F}{\partial \sigma_{ij}}} \tag{17}$$

The total strain increment can be obtained by the sum of the elastic strain increment and the plastic strain increment

$$d\varepsilon_{ij} = d\varepsilon_{ij}^e + d\varepsilon_{ij}^p \tag{18}$$

The elastic strain increment can be written as

$$d\varepsilon_{ij}^e = d\varepsilon_{ij} - d\varepsilon_{ij}^p \tag{19}$$

From Equations (8) and (19), we can get

$$d\sigma_{ij} = C_{ijkl}^e \left( d\varepsilon_{kl} - d\varepsilon_{kl}^p \right) \tag{20}$$

Then from Equations (14) and (20), we can get the expression of the stress increment as follows

$$d\sigma_{ij} = C_{ijkl}^e \left( d\varepsilon_{kl} - d\lambda \frac{\partial F}{\partial \sigma_{kl}} \right) \tag{21}$$

Substituting Equation (17) into Equation (21) results in the expression of the scalar multiplier $d\lambda$ as

$$d\lambda = \frac{\frac{\partial F}{\partial \sigma_{ij}} C_{ijkl}^e d\varepsilon_{ij}}{\left( \frac{\partial F}{\partial \sigma_{ij}} C_{ijkl}^e \frac{\partial F}{\partial \sigma_{kl}} - \frac{\partial F}{\partial K} \frac{\partial K}{\partial \varepsilon_{ij}^p} \frac{\partial F}{\partial \sigma_{ij}} \right)} \tag{22}$$

Then, from Equations (14) and (22), the plastic strain increment can be written as

$$d\varepsilon_{ij}^p = \frac{\frac{\partial F}{\partial \sigma_{ij}} C_{ijkl}^e \frac{\partial F}{\partial \sigma_{ij}}}{\left( \frac{\partial F}{\partial \sigma_{ij}} C_{ijkl}^e \frac{\partial F}{\partial \sigma_{kl}} - \frac{\partial F}{\partial K} \frac{\partial K}{\partial \varepsilon_{ij}^p} \frac{\partial F}{\partial \sigma_{ij}} \right)} d\varepsilon_{kl} \tag{23}$$

### 2.3. Damage Evolution Model

The damage variable evolution equation of the current constitutive model was developed based on the Najar damage theory [42]. In this theory, the damage process of material can be considered as the energy dissipation process. The energy dissipation of concrete under uniaxial compression is illustrated in Figure 1. The damage variable can be expressed as

$$D = \frac{W_0 - W_{PE}}{W_0} \tag{24}$$

where $W_0$ is the work corresponding to the strain $\varepsilon$ in the undamaged state, $W_0 = \frac{1}{2} E_0 \varepsilon^2$, and $E_0$ is the initial tangential modulus. $W_{PE}$ is the work corresponding to the strain $\varepsilon$ in the damaged state, $W_{PE} = W_P + W_E = \frac{1}{2} \sigma \varepsilon$; $W_E$ is the elastic strain energy and $W_P$ is the plastic strain energy. The damage variable is then written as

In the current study, the following damage variable is proposed to characterize the damage evolution of concrete under the uniaxial compression at different temperatures

$$D = M \left( \frac{\varepsilon}{\varepsilon_u^T} \right)^N \tag{26}$$

where $\varepsilon_u^T$ is the peak strain of concrete at temperature $T$, $\varepsilon_u^T / \varepsilon_u = 1 + \left( 1500T + 5T^2 \right) \times 10^{-6}$, and $\varepsilon_u$ is the peak strain of concrete at the normal temperature. The above damage variable evolution equation is expressed by the macro mechanical variables. Damage parameters $M$ and $N$ can be expressed as

$$M = M_1 \ln \left( \frac{f_c^T}{E_0 \left( \varepsilon_u^T \right)^2} \right) + M_2 \tag{27}$$

$$N = N_1 \ln \left( \frac{f_c^T}{E_0 \left( \varepsilon_u^T \right)^2} \right) + N_2 \tag{28}$$

where damage parameters $M_1$, $M_2$, $N_1$ and $N_2$ describe the damage evolution rate of concrete, and they can be demarcated from the uniaxial compression stress–strain curves of concrete at different temperatures.

$$D = 1 - \frac{\sigma}{E_0 \varepsilon} \tag{25}$$

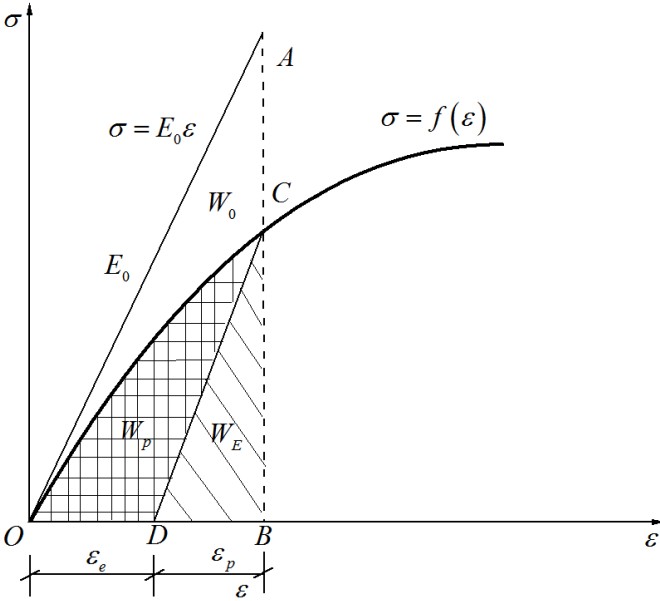

**Figure 1.** Energy dissipation of concrete under uniaxial compression [35].

### 2.4. Formulation of Developed Constitutive Model

During the elastic stage, no plastic strain arises, and the total strain increment is equal to the elastic strain increment. The stress increment can be expressed as

$$d\sigma_{ij} = C^e_{ijkl} d\varepsilon^e_{kl} = C^e_{ijkl} d\varepsilon_{kl} \tag{29}$$

The total stress increment vector can be written as

$$\{d\sigma\} = C^e_{ijkl}\{d\varepsilon\} \tag{30}$$

For the elastoplastic stage, plastic strain arises. The plastic strain increment is expressed as Equation (23), and the elastic strain increment can be written as follows

$$d\varepsilon^e_{ij} = d\varepsilon_{ij} - d\varepsilon^p_{ij} = d\varepsilon_{ij} - \frac{\frac{\partial F}{\partial \sigma_{ij}} C^e_{ijkl} d\varepsilon_{kl}}{\left(\frac{\partial F}{\partial \sigma_{ij}} C^e_{ijkl} \frac{\partial F}{\partial \sigma_{kl}} - \frac{\partial F}{\partial K} \frac{\partial K}{\partial \varepsilon^p_{ij}} \frac{\partial F}{\partial \sigma_{ij}}\right)} \frac{\partial F}{\partial \sigma_{ij}} \tag{31}$$

The incremental stress–strain relationship can be expressed as

$$d\sigma_{ij} = C^e_{ijkl} \left( d\varepsilon_{ij} - \frac{\frac{\partial F}{\partial \sigma_{ij}} C^e_{ijkl} d\varepsilon_{kl}}{\left(\frac{\partial F}{\partial \sigma_{ij}} C^e_{ijkl} \frac{\partial F}{\partial \sigma_{kl}} - \frac{\partial F}{\partial K} \frac{\partial K}{\partial \varepsilon^p_{ij}} \frac{\partial F}{\partial \sigma_{ij}}\right)} \frac{\partial F}{\partial \sigma_{kl}} \right) \tag{32}$$

The above equation is the general expression of the constitutive model proposed in the current study. Due to the complexity of the mechanical behaviors of concrete under high temperature conditions, it is not possible to present the stress–strain relationship in an explicit form, where all the model parameters are included. The deviatoric derivative

terms in Equation (32) need to be computed, and then the stress increment can be obtained using Equation (32).

The deviatoric derivative terms of the yield function to the stress components are written as follows

$$
\begin{aligned}
\frac{\partial F}{\partial \sigma_{11}} &= \alpha + \frac{2\sigma_{11}-\sigma_{22}-\sigma_{33}}{6\sqrt{\frac{1}{6}\left[(\sigma_{11}-\sigma_{22})^2+(\sigma_{22}-\sigma_{33})^2+(\sigma_{33}-\sigma_{11})^2\right]+\sigma_{12}^2+\sigma_{23}^2+\sigma_{31}^2}} \\
\frac{\partial F}{\partial \sigma_{22}} &= \alpha + \frac{2\sigma_{22}-\sigma_{11}-\sigma_{33}}{6\sqrt{\frac{1}{6}\left[(\sigma_{11}-\sigma_{22})^2+(\sigma_{22}-\sigma_{33})^2+(\sigma_{33}-\sigma_{11})^2\right]+\sigma_{12}^2+\sigma_{23}^2+\sigma_{31}^2}} \\
\frac{\partial F}{\partial \sigma_{33}} &= \alpha + \frac{2\sigma_{33}-\sigma_{22}-\sigma_{11}}{6\sqrt{\frac{1}{6}\left[(\sigma_{11}-\sigma_{22})^2+(\sigma_{22}-\sigma_{33})^2+(\sigma_{33}-\sigma_{11})^2\right]+\sigma_{12}^2+\sigma_{23}^2+\sigma_{31}^2}} \\
\frac{\partial F}{\partial \sigma_{12}} &= \frac{\sigma_{12}}{6\sqrt{\frac{1}{6}\left[(\sigma_{11}-\sigma_{22})^2+(\sigma_{22}-\sigma_{33})^2+(\sigma_{33}-\sigma_{11})^2\right]+\sigma_{12}^2+\sigma_{23}^2+\sigma_{31}^2}} \\
\frac{\partial F}{\partial \sigma_{23}} &= \frac{\sigma_{23}}{6\sqrt{\frac{1}{6}\left[(\sigma_{11}-\sigma_{22})^2+(\sigma_{22}-\sigma_{33})^2+(\sigma_{33}-\sigma_{11})^2\right]+\sigma_{12}^2+\sigma_{23}^2+\sigma_{31}^2}} \\
\frac{\partial F}{\partial \sigma_{31}} &= \frac{\sigma_{31}}{6\sqrt{\frac{1}{6}\left[(\sigma_{11}-\sigma_{22})^2+(\sigma_{22}-\sigma_{33})^2+(\sigma_{33}-\sigma_{11})^2\right]+\sigma_{12}^2+\sigma_{23}^2+\sigma_{31}^2}}
\end{aligned}
\tag{33}
$$

The deviatoric derivative terms of the hardening parameter to the strain components are written as follows

$$
\begin{aligned}
\frac{\partial K}{\partial \varepsilon_{11}^{p}} &= -\frac{\sqrt{2}}{3}\frac{A}{\left(A+B\varepsilon_d^p\right)^2}\frac{2\varepsilon_{11}^p-\varepsilon_{22}^p-\varepsilon_{33}^p}{\sqrt{(\varepsilon_{11}^p-\varepsilon_{22}^p)^2+(\varepsilon_{22}^p-\varepsilon_{33}^p)^2+(\varepsilon_{33}^p-\varepsilon_{11}^p)^2+6(\varepsilon_{12}^{p2}+\varepsilon_{13}^{p2}+\varepsilon_{23}^{p2})}} \\
\frac{\partial K}{\partial \varepsilon_{22}^{p}} &= -\frac{\sqrt{2}}{3}\frac{A}{\left(A+B\varepsilon_d^p\right)^2}\frac{2\varepsilon_{22}^p-\varepsilon_{11}^p-\varepsilon_{33}^p}{\sqrt{(\varepsilon_{11}^p-\varepsilon_{22}^p)^2+(\varepsilon_{22}^p-\varepsilon_{33}^p)^2+(\varepsilon_{33}^p-\varepsilon_{11}^p)^2+6(\varepsilon_{12}^{p2}+\varepsilon_{13}^{p2}+\varepsilon_{23}^{p2})}} \\
\frac{\partial K}{\partial \varepsilon_{33}^{p}} &= -\frac{\sqrt{2}}{3}\frac{A}{\left(A+B\varepsilon_d^p\right)^2}\frac{2\varepsilon_{33}^p-\varepsilon_{11}^p-\varepsilon_{22}^p}{\sqrt{(\varepsilon_{11}^p-\varepsilon_{22}^p)^2+(\varepsilon_{22}^p-\varepsilon_{33}^p)^2+(\varepsilon_{33}^p-\varepsilon_{11}^p)^2+6(\varepsilon_{12}^{p2}+\varepsilon_{13}^{p2}+\varepsilon_{23}^{p2})}} \\
\frac{\partial K}{\partial \varepsilon_{12}^{p}} &= -2\sqrt{2}\frac{A}{\left(A+B\varepsilon_d^p\right)^2}\frac{\varepsilon_{12}^p}{\sqrt{(\varepsilon_{11}^p-\varepsilon_{22}^p)^2+(\varepsilon_{22}^p-\varepsilon_{33}^p)^2+(\varepsilon_{33}^p-\varepsilon_{11}^p)^2+6(\varepsilon_{12}^{p2}+\varepsilon_{13}^{p2}+\varepsilon_{23}^{p2})}} \\
\frac{\partial K}{\partial \varepsilon_{23}^{p}} &= -2\sqrt{2}\frac{A}{\left(A+B\varepsilon_d^p\right)^2}\frac{\varepsilon_{23}^p}{\sqrt{(\varepsilon_{11}^p-\varepsilon_{22}^p)^2+(\varepsilon_{22}^p-\varepsilon_{33}^p)^2+(\varepsilon_{33}^p-\varepsilon_{11}^p)^2+6(\varepsilon_{12}^{p2}+\varepsilon_{13}^{p2}+\varepsilon_{23}^{p2})}} \\
\frac{\partial K}{\partial \varepsilon_{31}^{p}} &= -2\sqrt{2}\frac{A}{\left(A+B\varepsilon_d^p\right)^2}\frac{\varepsilon_{31}^p}{\sqrt{(\varepsilon_{11}^p-\varepsilon_{22}^p)^2+(\varepsilon_{22}^p-\varepsilon_{33}^p)^2+(\varepsilon_{33}^p-\varepsilon_{11}^p)^2+6(\varepsilon_{12}^{p2}+\varepsilon_{13}^{p2}+\varepsilon_{23}^{p2})}}
\end{aligned}
\tag{34}
$$

The thermomechanical coupling constitutive model of concrete developed in the current study includes nine parameters, which are the elastic modulus $E$; Poisson's ratio $v$; cylinder compressive strength $f_c$ and peak strain $\varepsilon_u$ at the normal temperature, which are referred to the unconfined concrete; Drucker–Prager strength criterion parameter $\alpha$ and damage parameters $M_1$, $M_2$, $N_1$ and $N_2$. The parameters $E$, $v$, $f_c$ and $\varepsilon_u$ can be measured by the conventional physical and mechanical tests of concrete. The parameter $\alpha$ can take a value of 0.01. The damage parameters $M_1$, $M_2$, $N_1$ and $N_2$ can be demarcated from the uniaxial compression stress–strain curves of concrete at different temperatures.

## 3. Model Validation

In this section, the developed constitutive model is validated by the comparison of the numerical simulation results of the model with the test results. The developed constitutive model was coded and implemented into the finite element software ABAQUS through the UMAT interface. The uniaxial compression tests of concrete at different temperatures were numerically simulated in ABAQUS, and the results are compared with the test results [6]. In these tests, a high temperature heating furnace was assembled with the compression testing machine. The tested concrete specimens were placed in the furnace chamber and heated to the specified temperature. This temperature was maintained for six hours to ensure the specimen was heated evenly and attain the specified temperature. Stainless steel plates with outward-extending poles were installed on both ends of specimen. The outward extending poles were outside the heating furnace and connected to the displacement

transducer to measure the deformation of the specimen. At a specified test temperature, the compressive load was applied to the tested concrete specimen. The corresponding stress–strain curve was obtained with the X–Y function recorder.

The calculation model for the uniaxial compression of a concrete cubic element is shown in Figure 2. The concrete element was simulated with the C3D8 element with eight nodes in ABAQUS. The side length of the element is 1 mm. The bottom surface of the element is constrained, and vertical displacement is applied to the top surface to simulate the uniaxial compression test. The numerical simulation of the uniaxial compression tests of concrete at different temperatures were implemented with the thermodynamic properties of concrete being characterized by the proposed thermomechanical coupling constitutive model. The concrete strength is C20 and C40, and the corresponding model parameters are listed in Tables 1 and 2, respectively. $E_{(T)}$ is the elastic modulus of concrete corresponding to the temperature $T$.

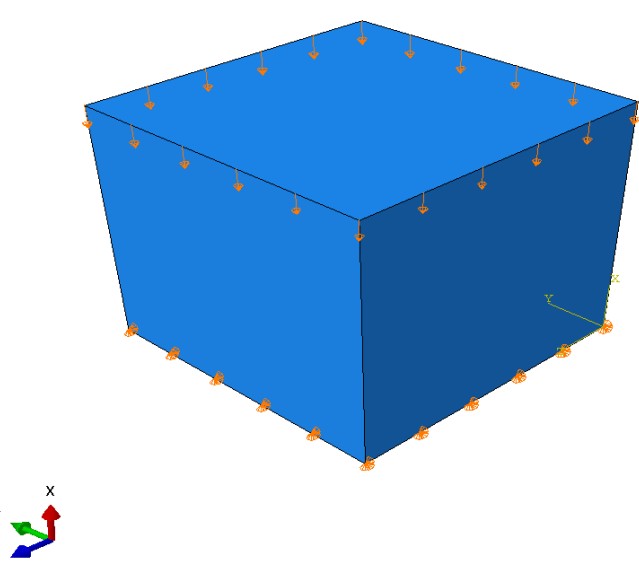

**Figure 2.** Calculation model for uniaxial compression of the concrete cubic element.

**Table 1.** Constitutive model parameters for C20 concrete.

| $E_{(20)}$ (GPa) | $E_{(100)}$ (GPa) | $E_{(300)}$ (GPa) | $E_{(500)}$ (GPa) | $E_{(700)}$ (GPa) | $v$ | $\alpha$ | $\varepsilon_u$ | $f_c$ (MPa) | $M_1$ | $M_2$ | $N_1$ | $N_2$ |
|---|---|---|---|---|---|---|---|---|---|---|---|---|
| 21.781 | 18.732 | 11.696 | 5.793 | 1.481 | 0.15 | 0.01 | 0.002 | 20.435 | 0.0189 | 0.9102 | 0.0296 | 0.6225 |

**Table 2.** Constitutive model parameters for C40 concrete.

| $E_{(20)}$ (GPa) | $E_{(100)}$ (GPa) | $E_{(300)}$ (GPa) | $E_{(500)}$ (GPa) | $E_{(700)}$ (GPa) | $v$ | $\alpha$ | $\varepsilon_u$ | $f_c$ (MPa) | $M_1$ | $M_2$ | $N_1$ | $N_2$ |
|---|---|---|---|---|---|---|---|---|---|---|---|---|
| 39.278 | 33.779 | 21.090 | 10.446 | 2.671 | 0.15 | 0.01 | 0.002 | 36.85 | 0.0189 | 0.9102 | 0.0296 | 0.6225 |

The comparison of the numerical results of the stress–strain relationship simulated by the developed constitutive model with the test results is shown in Figures 3 and 4. It is illustrated that the simulation results accord well with the test results at different temperatures. The simulation results show that the stress–strain curves tend to grow more gently with the rise of temperature. With the rise of temperature, the peak stress of concrete decreases, but the strain corresponding to the peak stress increases. This accords with the essential mechanical properties of concrete at different temperatures. The above comparison indicates that the constitutive model developed in the current study

can characterize the mechanical performance of concrete at different temperatures with considerable accuracy.

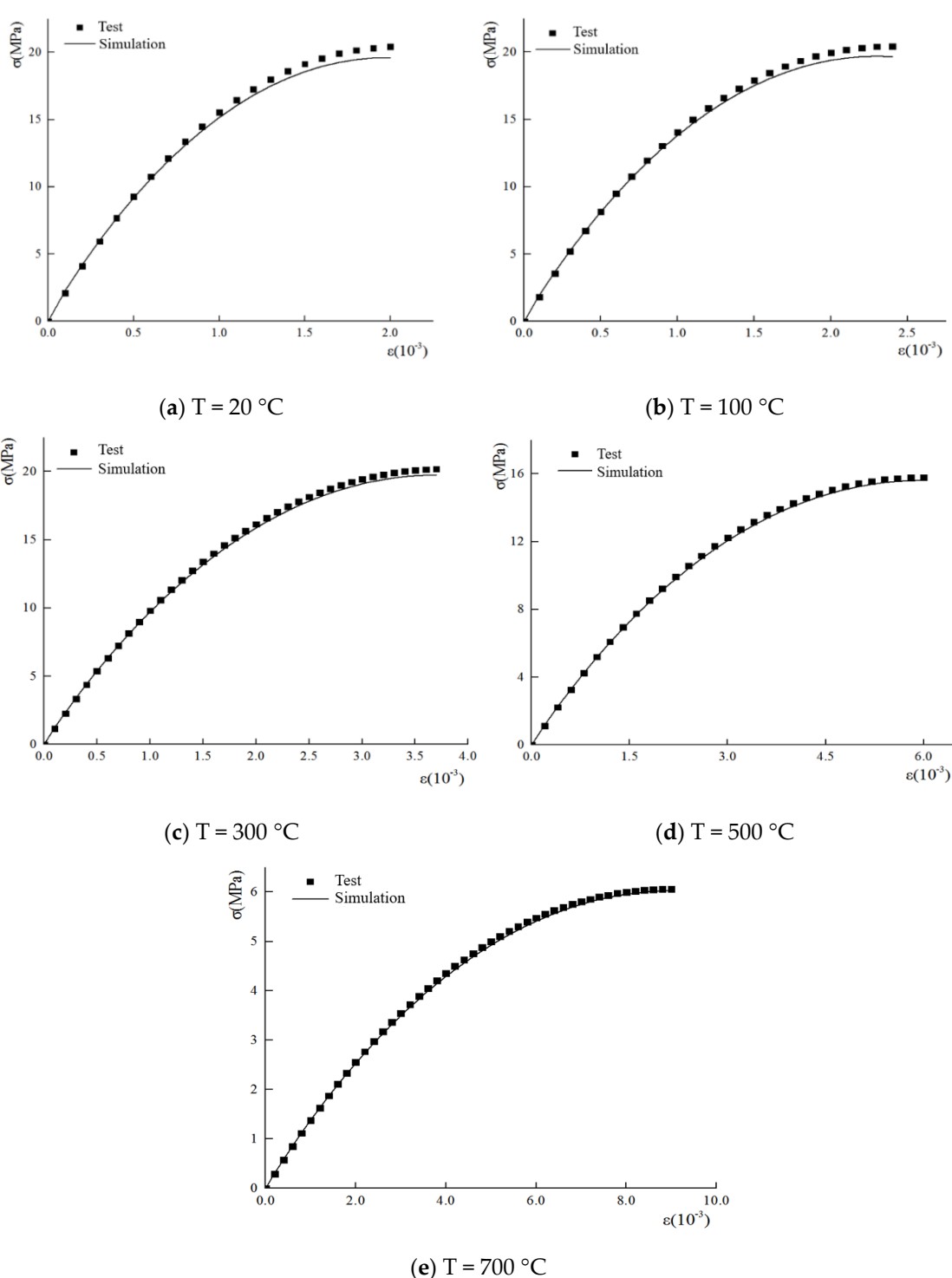

**Figure 3.** Comparison of simulation results and test results for C20 concrete.

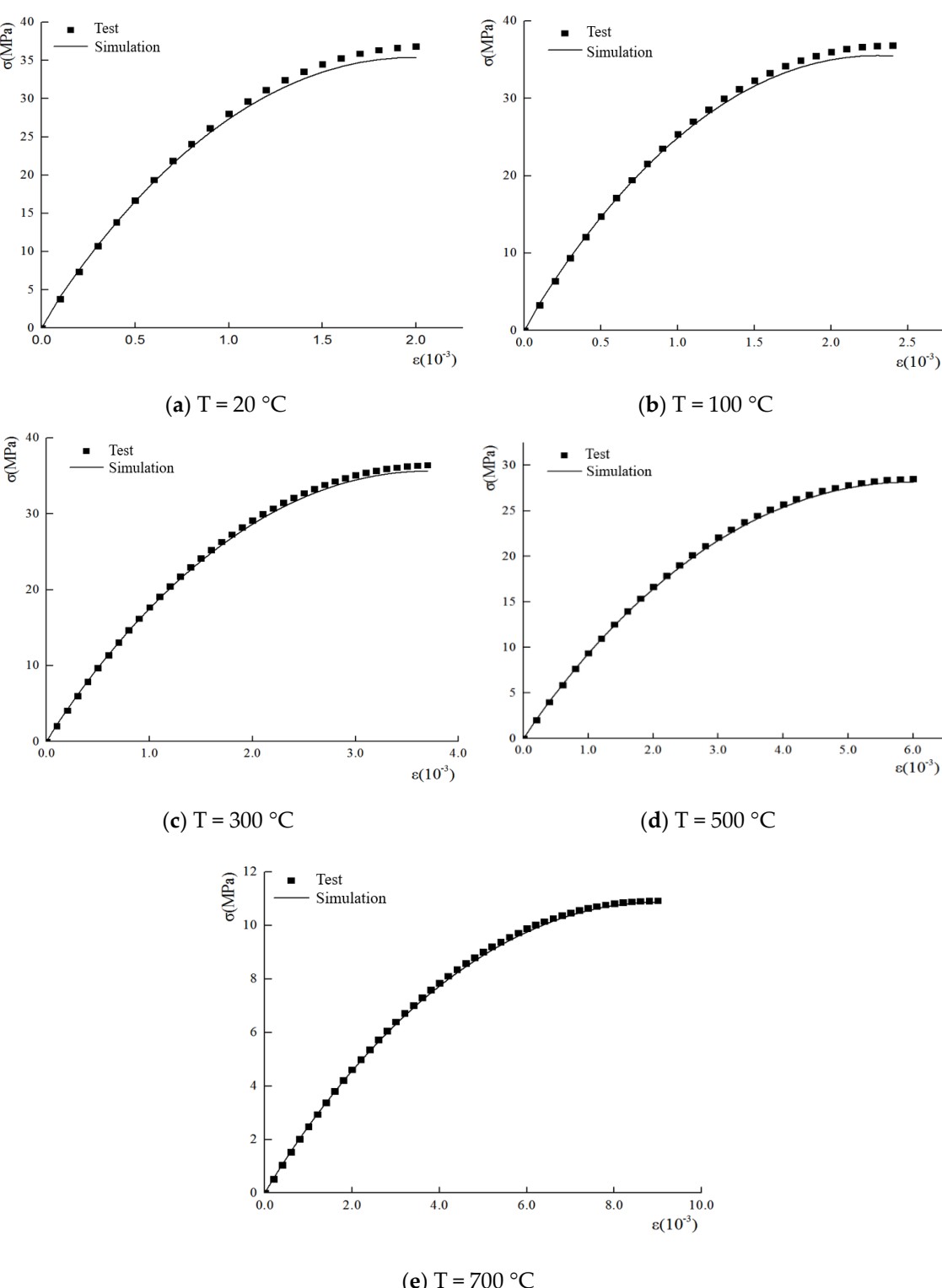

**Figure 4.** Comparison of simulation results and test results for C40 concrete.

It is noted that the constitutive model proposed in the current study can only simulate the hardening segments or ascending segments of the stress–strain relationship curves of concrete at different temperatures. It cannot simulate the softening segments or descending segments of the stress–strain relationship curves. In Figures 3 and 4, only the test and simulation results of the ascending segments of the stress–strain relationship curves are presented for the purpose of comparison. The test results of the descending segments of

the stress–strain relationship curves are not presented, although they have been revealed in the original literature.

The relationship between the damage variable value and the compressive strain at different temperatures for C20 and C40 concrete is presented in Figures 5 and 6. It is shown that the damage of concrete increases with the increase of compressive strain at a given temperature. On the other hand, the damage growth rate decreases with the rise of temperature. This indicates that the rise of temperature reduces the brittleness of concrete while improve its toughness.

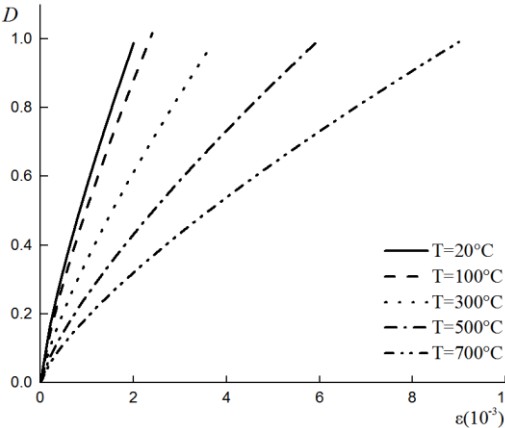

**Figure 5.** Damage variable vs. strain at different temperatures (C20 concrete).

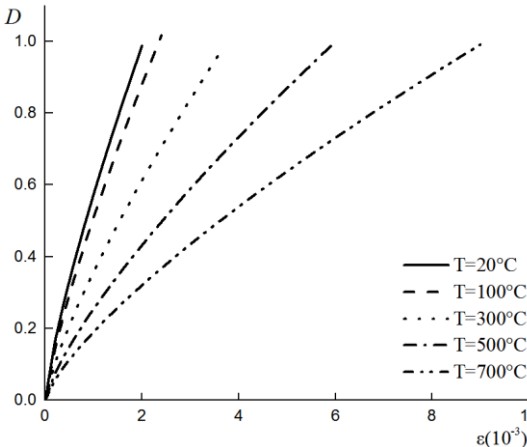

**Figure 6.** Damage variable vs. strain at different temperatures (C40 concrete).

## 4. Model Application

The proposed constitutive model was applied as the material model of concrete to simulate an axially compressive concrete column. The simulated concrete column is 3 m long. It has a square cross section with a side length of 500 mm. The bottom surface of the concrete column is fixed, and all the degrees of freedom are constrained. The top surface of the concrete column is subjected to an axial uniform load along the vertically downward direction to simulate the axially compressive loading case. The concrete strength is C40. The finite element discretization of column was implemented with the C3D8 element in ABAQUS. The element size is $100 \times 100 \times 100$ mm. The calculation model consists of 750 elements and 1116 nodes. The total number of the degrees of freedom of the calculation model is 3240. The finite element calculation model is shown in Figure 7. The constitutive model parameters for the simulation are listed in Table 3.

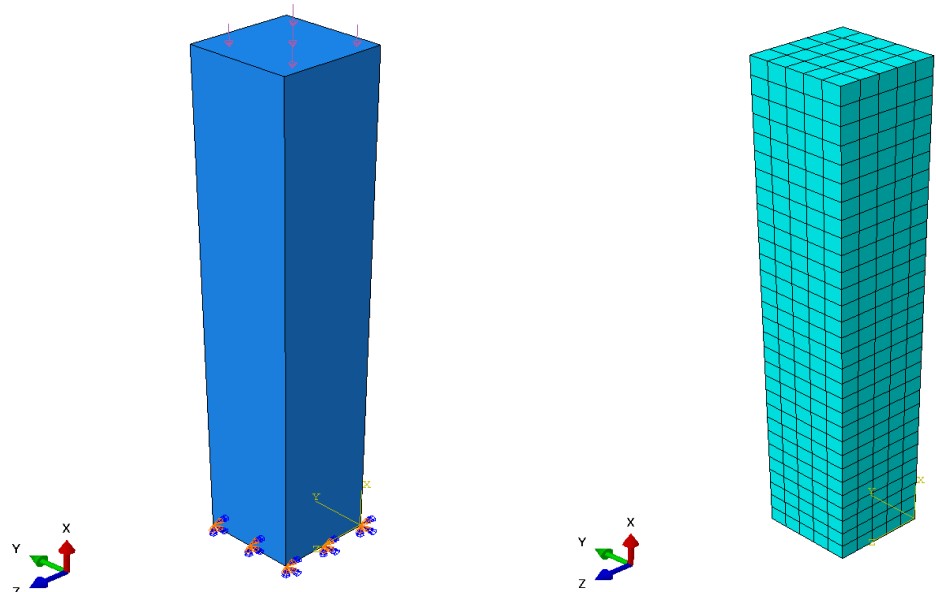

(**a**) Constraint condition and load.　　　　　(**b**) Finite element meshes.

**Figure 7.** Finite element calculation model of axially compressive concrete column.

**Table 3.** Constitutive model parameters for the simulation.

| | $E_{(T)}$ (GPa) | $v$ | $\alpha$ | $\varepsilon_u$ | $f_c$ (MPa) | $M_1$ | $M_2$ | $N_1$ | $N_2$ |
|---|---|---|---|---|---|---|---|---|---|
| $E_{(20)}$ | 39.278 | | | | | | | | |
| $E_{(100)}$ | 33.779 | | | | | | | | |
| $E_{(200)}$ | 27.001 | | | | | | | | |
| $E_{(300)}$ | 21.090 | 0.15 | 0.01 | 0.002 | 36.85 | 0.0189 | 0.9102 | 0.0296 | 0.6225 |
| $E_{(400)}$ | 15.751 | | | | | | | | |
| $E_{(500)}$ | 10.446 | | | | | | | | |
| $E_{(600)}$ | 5.725 | | | | | | | | |
| $E_{(700)}$ | 2.671 | | | | | | | | |

The vertical displacement nephograms of the simulated axially compressive concrete column at different temperatures are shown in Figure 8. The comparison of the load–displacement relationship of the simulated column at different temperatures is presented in Figure 9. It is indicated that under the action of the load with the same amplitude, the vertical deformation of the concrete column increases with the rise of temperature, especially when the temperature is higher than 400 °C. This results from the degradation of the mechanical performance of concrete at elevated temperatures. The above results simulated by the proposed constitutive model are consistent with the essential mechanical response behaviors of concrete members at different temperatures.

The damage nephograms of the axially compressive concrete column at different temperatures are illustrated in Figure 10. It is shown that more serious damage to the concrete column can be observed with the rise of temperature, especially when the temperature is higher than 400 °C. This also characterizes the degradation of concrete strength at elevated temperatures. The above simulation results indicate that the developed constitutive model in the current study can be applied to the numerical simulation of the mechanical behaviors of concrete members at high temperatures.

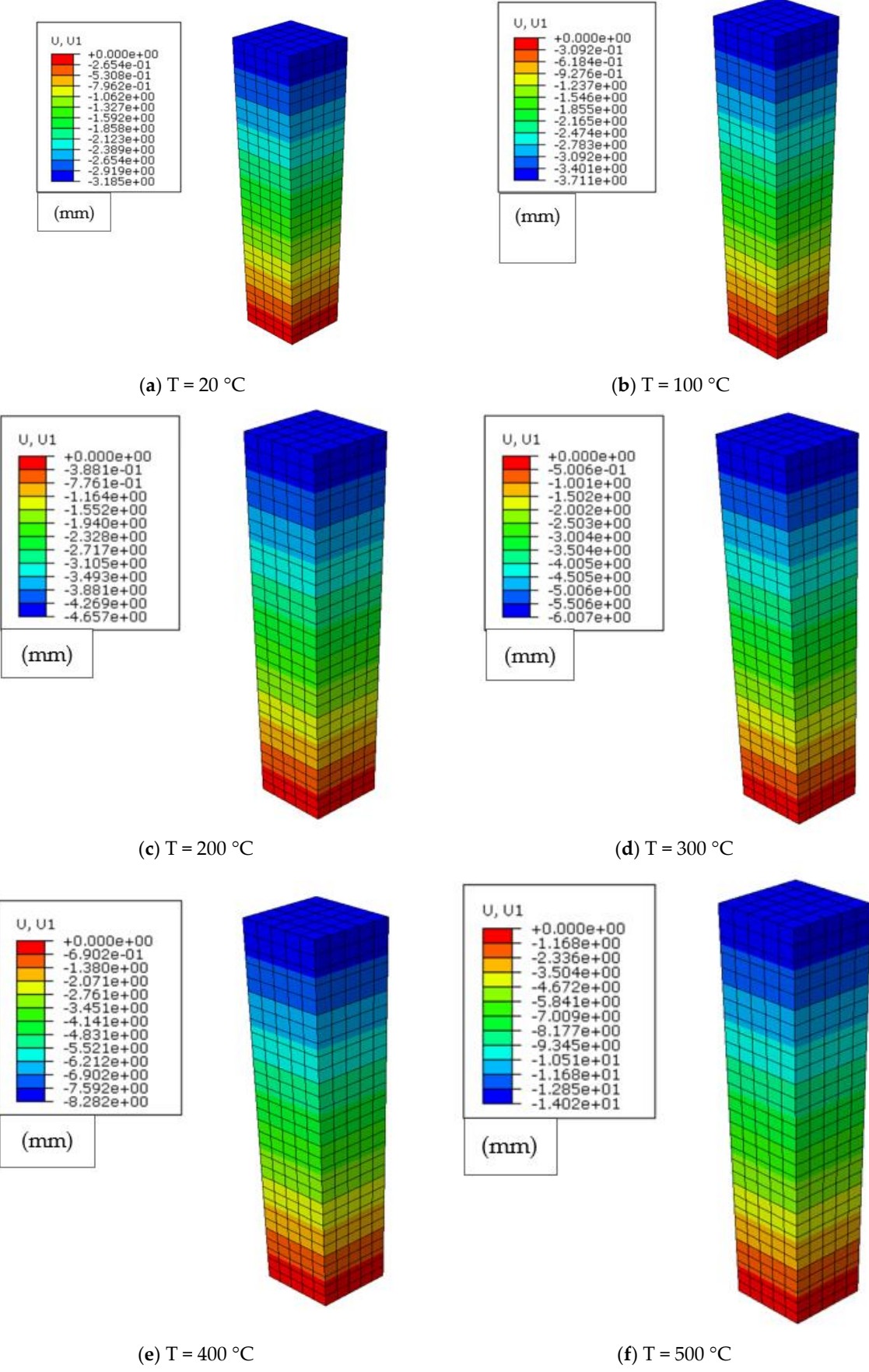

(**a**) T = 20 °C

(**b**) T = 100 °C

(**c**) T = 200 °C

(**d**) T = 300 °C

(**e**) T = 400 °C

(**f**) T = 500 °C

**Figure 8.** *Cont.*

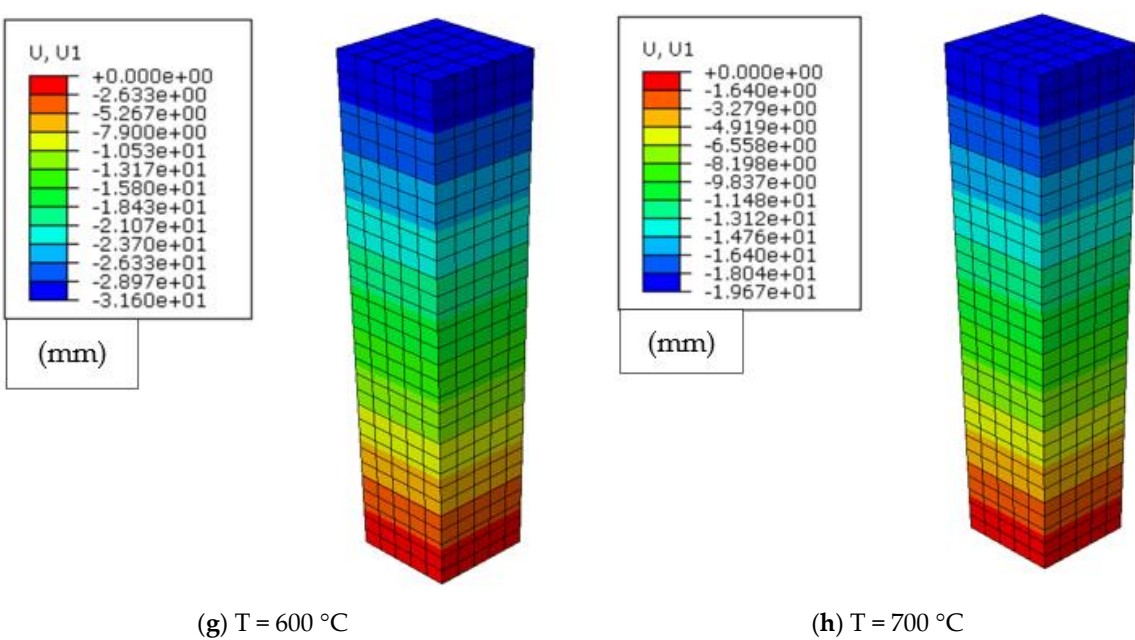

(**g**) T = 600 °C                                 (**h**) T = 700 °C

**Figure 8.** Vertical displacement nephograms of axially compressive concrete column at different temperatures.

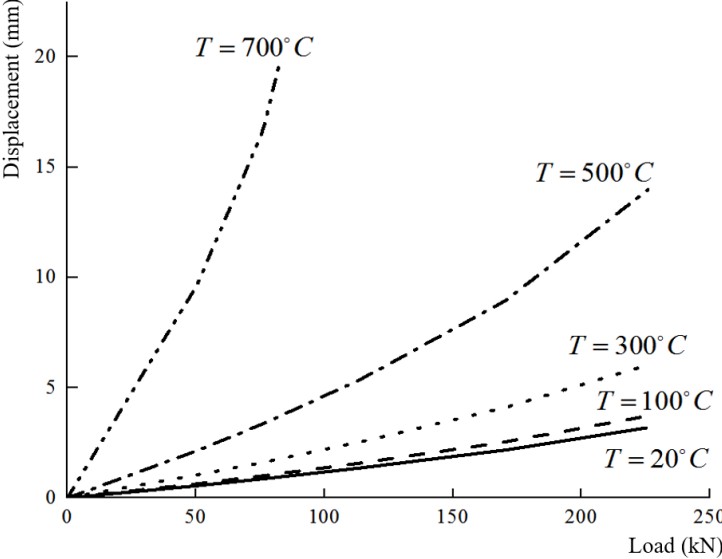

**Figure 9.** Comparison of load–displacement relationship of axially compressive concrete column at different temperatures.

In the current numerical simulation, only the mechanical behaviors of a plain concrete column at different temperatures were investigated, but the practical structural columns have longitudinal and transverse reinforcements, which have an important effect on their mechanical behaviors. The interaction between the concrete and reinforcements and its variation with the temperature are not included in the proposed constitutive model and current numerical simulation. They should be taken into account in future investigation.

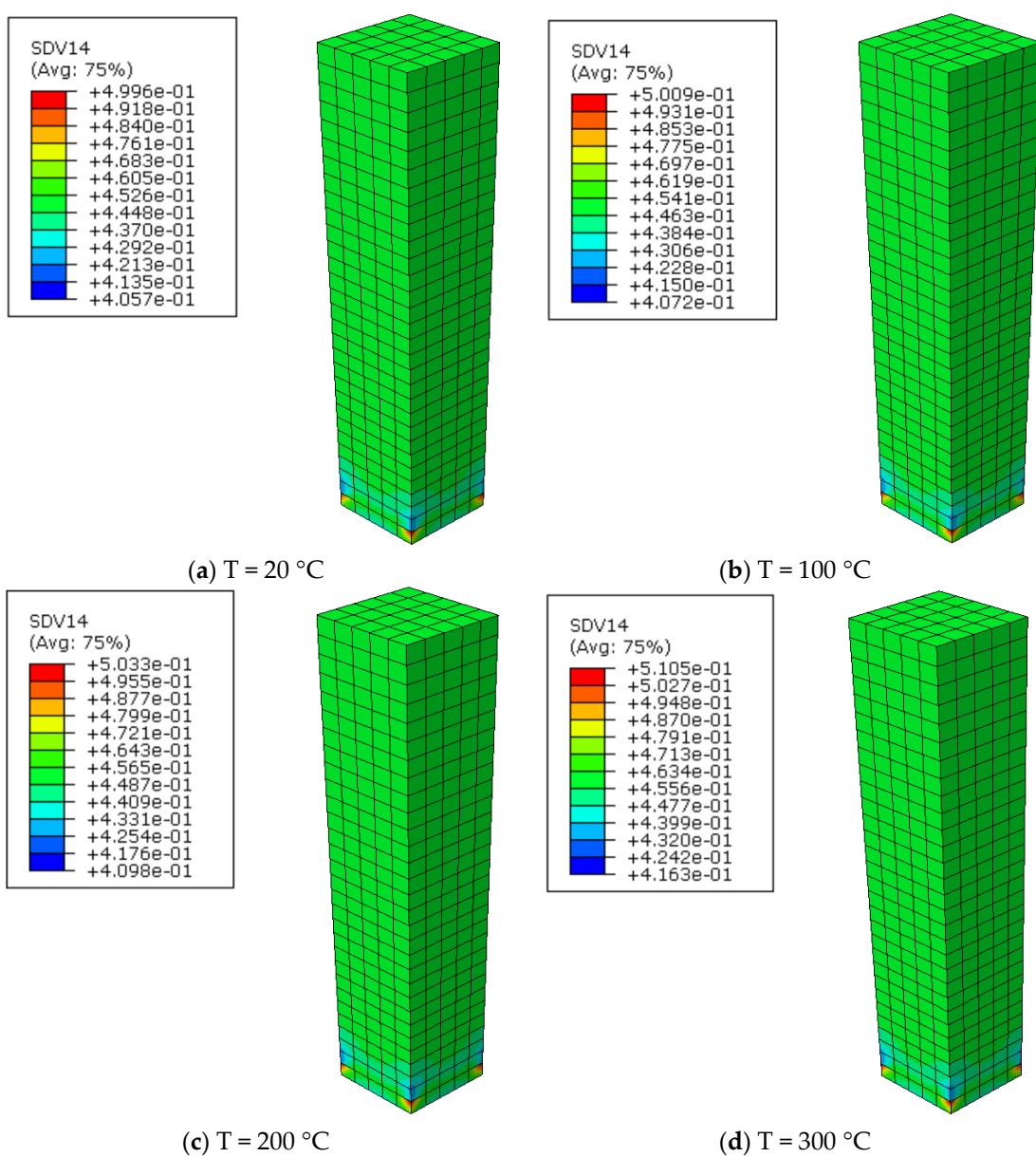

(**a**) T = 20 °C

(**b**) T = 100 °C

(**c**) T = 200 °C

(**d**) T = 300 °C

**Figure 10.** *Cont.*

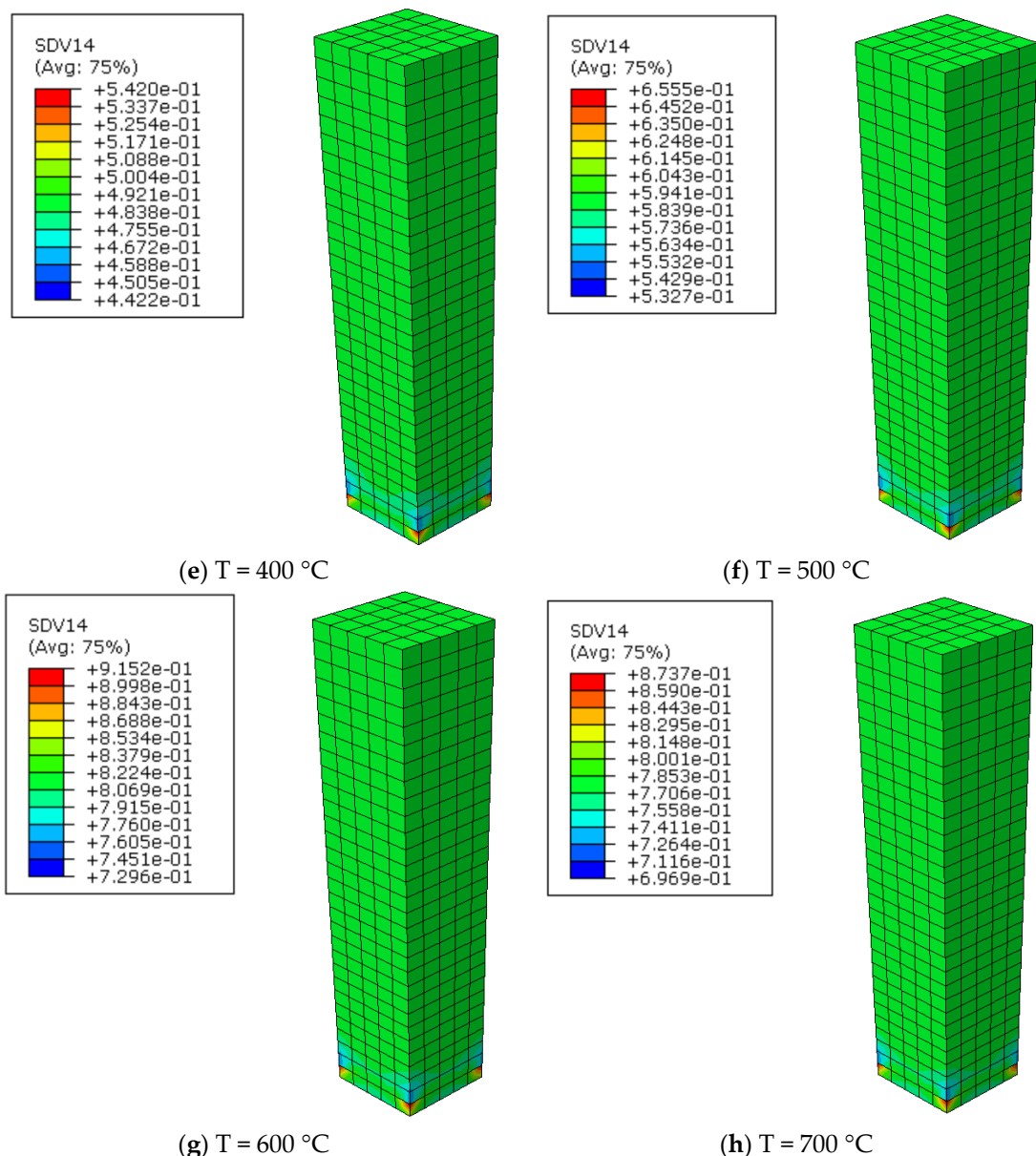

**Figure 10.** Damage nephograms of axially compressive concrete column at different temperatures.

## 5. Conclusions

A thermomechanical coupling constitutive model for the description of the mechanical behaviors of concrete at different temperatures is proposed. The expression of the elastic strain increment is derived with the free energy function including the temperature variable. The expression of the plastic strain increment is derived from the yield function based on the Drucker–Prager strength criterion. The elastoplastic damage effect is included in this constitutive model. The damage variable is included in the yield function to consider the effect of the damage on the elastoplastic mechanical behaviors of concrete. The damage variable evolution equation is expressed by the macro mechanical variables.

The proposed constitutive model was coded and implemented into the finite element software ABAQUS through UMAT interface. The uniaxial compression tests of concrete at different temperatures were numerically simulated, and the results are compared with the test results to validate the proposed constitutive model. The simulation results accord well with the test results at different temperatures. This indicates that the proposed constitutive model can characterize the mechanical behaviors of concrete at different temperatures with

considerable accuracy. The proposed constitutive model was then applied to simulate an axially compressive concrete column. The simulation results are consistent with the essential mechanical response behaviors of concrete members at different temperatures.

The constitutive model proposed in the current study has relatively simple expressions, and it is convenient for the computer implementation of this model. Fewer model parameters are included in this model, and they all have explicit physical meanings. It can be applied to the numerical simulation of the mechanical performance of concrete members at different temperatures.

The presented constitutive model has some vital disadvantages and limitations. It can only be applied to the uniaxial compression loading case and can only simulate the ascending segment of the stress–strain relationship of concrete. Some important factors, such as moisture, pore pressure, chemical action and loading rate, are not included in this model. On the other hand, the presented model was developed mainly based on the elastoplastic theory and damage mechanics. The constitutive model strictly based on inversible thermodynamics should be developed in future study.

The proposed constitutive model in the current study cannot be applied to confined concrete, which is the actual working state of the concrete in structural members. The presented model should be revised and developed for application to confined concrete in future investigation.

**Author Contributions:** L.L. developed the thermomechanical coupling constitutive model and performed the formula derivation of the model. L.L. prepared the manuscript. H.W. performed the program composition and numerical simulation for the model validation. J.W. discussed the numerical results and provided suggestions on the overall organization of the paper. W.J. performed the numerical simulation for the model application and revised the manuscript. All authors have read and agreed to the published version of the manuscript.

**Funding:** This research was funded by the National Natural Science Foundation of People's Republic of China [No. 52078288] and the Beijing Municipal Natural Science Foundation [No. 8212001].

**Informed Consent Statement:** Not applicable.

**Data Availability Statement:** The data presented in this study are available on request from the corresponding author. The data are not publicly available due to the intellectual property.

**Conflicts of Interest:** The authors declare no conflict of interest.

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
