# Peer review of "A Thermomechanical Coupling Constitutive Model of Concrete Including Elastoplastic Damage"

_applsci, doi:10.3390/app11020604_

Round 1

Reviewer 1 Report

Dear Authors,

I read your contribution with interest and attention. I can surely say that there are several remarkable points for reflection and, in general, that the manuscript is valid. At the same time, I have to say that there are several issues that require correction or explanation.

Kind regards

  1. The authors are to be commended for a thorough literature review, although it seems that the review should be supplemented with more articles from recent years (2018-2020).
  2. The novelty of the work should be clearly indicated in the introduction.
  3. Please provide the description of the manuscript structure. Provide a brief description of the content of each section (see 10.3390/ma13071630).
  4. The authors refer to the Drucker-Prager constitutive model on several occasions in the article; however, there is no proper literature reference. Please add.
  5. The authors are to be commended for a detailed presentation of finite element model; however, I believe that adding an additional information about number of model degrees of freedom and an exact model of material used would be advisable. Additionally, a detailed information about boundary conditions and applied loads is needed.
  6. I believe that Fig. 8 in current is redundant since Fig. 9 presents a full characteristic. I suggest showing only some representative cases in Fig. 8. The same with Fig. 10.
  7. Since the submitted manuscript does not contain the results of experimental tests, it seems that the modeling results should be compared not only with the finite element model, but also with other models known from the literature. If this is not possible, please discuss the presented model and obtained results in relation to the models known from the literature (indicate the advantages, disadvantages, and limitations of the developed model in relation to the others known from the literature, please see Section 4 in 10.3390/ma13143151).

Reviewer 2 Report

In the paper a coupled constitutive model for concrete, capable of predicting the thermal and mechanical behavior is presented. Firstly, the analytical formulation is presented. Then, validations with experimental results are illustrated and commented. The paper is well-structured and its contents are clear and worth to note, because it focuses on a coupled model where also temperature dependence is taken into account for describing the mechanical behavior of concrete. Therefore, it is opinion of the Reviewer that the manuscript may be published after a revision following the comments/suggestions below listed.

Point 1

It is well known in literature that the compressed concrete is heavily conditioned by passive confinement offered by transverse reinforcements, consisting of internal hoops and additional external strengthenings, such as FRP wraps/steel jackets. This aspect should be at least clearly mentioned in the introduction considering, among the others, the following works:

  • Laterza, M., D'Amato, M., Thanthirige, L.P., Braga, F., Gigliotti, R. 2014. Comparisons of codal detailing rules for curvature ductility and numerical investigations. Open Construction and Building Technology Journal, 8 (1), pp. 132-141.
  • Mander JB, Priestley MJN, Park R.1988. Theoretical stress-strain model for confined concrete. J Struct Eng ASCE 1988;114(8):1804–26

Then, it should be clearly indicated if the model proposed regards also confined concrete. Otherwise, it is suggested of discussing of this in the conclusion section as future development. This is an important aspect since it permits of collocating the work presented in the current scientific literature.

Moreover, it should be specified if the cylinder compressive strength and related strain are referred to unconfined or confined concrete (line 237-238).

Please, if available provide a reference for the parameter alfa.

Point 2

The Reviewer does not understand if there is (or not) a possibility to report the sigma-epsilon curve in an explicit form, where all parameters are included. Please, comment this in the manuscript.

Point 3

Validations are conducted by referring to some experimental tests. Please provide more details about these tests and on how results were measured.

Point 4

Why is a softening branch not reported in all stress-strain curves of Figure 4 and Figure 5? It seems that in all tests the measurements are conducted until the peak strength. Please comment.

Point 5

On the simulated behavior of the concrete column. The Reviewer appreciates the numerical investigations. However, in the practice concrete columns have longitudinal and transverse reinforcements that will interact with the concrete, due to the temperature variation. At least, this aspect should be duly commented. Please add a measure units in Figure 8 and Figure 10.

Round 2

Reviewer 1 Report

I recommend manuscript for publication 

Reviewer 2 Report

The paper has been revised following the comments/suggestions provided. Therefore, it may be published in the present form.